# Progress in Research on Key Factors Regulating Lactation Initiation in the Mammary Glands of Dairy Cows

**DOI:** 10.3390/genes14061163

**Published:** 2023-05-26

**Authors:** Haoyue Guo, Jianyuan Li, Yuhao Wang, Xiang Cao, Xiaoyang Lv, Zhangping Yang, Zhi Chen

**Affiliations:** 1College of Animal Science and Technology, Yangzhou University, Yangzhou 225009, China; guohaoyue1001@163.com (H.G.); wangyuhao000701@163.com (Y.W.); mx120210843@yzu.edu.cn (X.C.); yzp@yzu.edu.cn (Z.Y.); 2Huanshan Group, Qingdao 266000, China; lijy163@vanke.com; 3Joint International Research Laboratory of Agriculture & Agri-Product Safety, Ministry of Education, Yangzhou University, Yangzhou 225009, China; dx120170085@yzu.edu.cn; 4International Joint Research Laboratory in Universities of Jiangsu Province of China for Domestic Animal Germplasm Resources and Genetic Improvement, Yangzhou 225009, China

**Keywords:** lactogenesis, mammary gland, dairy cows, hormonal

## Abstract

Lactation initiation refers to a functional change in the mammary organ from a non-lactating state to a lactating state, and a series of cytological changes in the mammary epithelium from a non-secreting state to a secreting state. Like the development of the mammary gland, it is regulated by many factors (including hormones, cytokines, signaling molecules, and proteases). In most non-pregnant animals, a certain degree of lactation also occurs after exposure to specific stimuli, promoting the development of their mammary glands. These specific stimuli can be divided into two categories: before and after parturition. The former inhibits lactation and decreases activity, and the latter promotes lactation and increases activity. Here we present a review of recent progress in research on the key factors of lactation initiation to provide a powerful rationale for the study of the lactation initiation process and mammary gland development.

## 1. Introduction

The mammary gland is an organ that is unique to mammals and serves a lactational function. Unlike other organs, most mammary gland tissue is developed after birth. Mammary gland development, an important indicator of a cow’s growth and development, is primarily determined by the number of lactating cells and the cow’s milk production capacity [1]. The reproduction of mammals usually involves multiple lactation cycles, which, in cows, can be divided into four stages based on their physiological characteristics and lactation levels: early lactation, middle lactation, late lactation, and dry lactation [2]. Furthermore, the endocrine system greatly influences the regulation of mammary gland development, lactation initiation, and lactation maintenance. Many factors (including hormones) act on target cells through cellular signal transduction pathways [3]. Therefore, relevant transcription factors are closely associated with the development of the mammary gland and the regulation of lactation [4]. Somatotropin and prolactin (PRL) have been found to be primary factors involved in the latter [5]. Interestingly, not all growth factors and hormones regulate mammary gland follicle development [6]. Lactation in dairy cows has been extensively revealed to be regulated by hormones in the neuroendocrine system, the autocrine and paracrine growth factors of the organism, and other tissues [7]. Furthermore, these hormones coordinate and synergize collaboratively to complete the development of the mammary gland and regulate lactation through endocrine, paracrine, and autocrine mechanisms [8].

The regulation of the lactation cycle plays a decisive role in improving milk production and optimizing milk quality [9]. It stimulates lactation initiation by promoting the proliferation and differentiation of mammary cells and the formation of a complete mammary gland structure [10]. During lactation, various external and internal factors influence the changes in milk composition and lactation volume. At the end of lactation, the mammary tissue gradually enters the dry lactation phase, where it undergoes repair and renewal in preparation for the next cycle of lactation [11]. It is therefore important to understand the changes in the lactation cycle of dairy cows and to tailor different lactation cycles accordingly, thereby effectively improving milk yield and milk quality. In addition to growth factors, many factors (including cytokines, signaling molecules, and proteases) are pivotal to the efficiency of substance metabolism, transformation, and uptake in cows. This, in turn, plays a significant role in the hormonal regulation of lactation [12], and also substantially impacts the structural development of the mammary gland and the cow’s lactation performance. This review synthesizes recent research progress on many key lactation factors and provides a powerful rationale for the study of lactation initiation. It is mainly divided into four points, including the hormonal regulation of lactation in dairy cows, the regulation of lactogenesis by cytokines and signaling molecules in dairy cows, the regulation of lactogenesis by proteases in dairy cows, and the regulation of lactogenesis by other factors in dairy cows.

## 2. Hormonal Regulation of Lactation in Dairy Cows

### 2.1. PRL

In addition to primarily promoting somatic cell proliferation and division, PRL regulates the development of the mammary gland, milk initiation, and lactation maintenance. It is therefore closely associated with milk yield and milk fat production [13]. To achieve lactation initiation, PRL initially exercises its biological function by binding to the PRL receptor on the cell surface of the target site to activate the JAK2 (tyrosine-protein kinase JAK2)/STAT5 (signal transducer and activator of transcription 5) signaling pathway; this is followed by the trans-acting factor STAT5 acting on the target sequence of the promoter region of the milk protein gene. PRL receptors are expressed in cows in both short and long forms, and their imbalanced ratios could cause decreased lactation and diseases to occur. Studies have shown that PRL plays a crucial role in lactation initiation, and its secretion increases prior to parturition. Furthermore, PRL, PRL receptors, and milk production have been reported to be significantly positively correlated, further indicating a relationship between PRL and lactation in dairy cows [14]. A different study has found that PRL promotes the proliferation of mammary epithelial cells in dairy cows [15], while its effect on their apoptosis is not significant. The possible underlying mechanism involves the binding of PRL to its receptors during mammary gland development, stimulating the development of mammary gland follicles, and subsequently stimulating the initiation and maintenance of lactation [16]. Therefore, it is important to explore the effect of PRL on the proliferation and apoptosis of mammary epithelial cells to understand lactation initiation and maintenance in dairy cows.

### 2.2. Estrogen

Estrogen is a major driver of breast remodeling and functional regulation. Among the diverse hormones involved in lactation regulation, estradiol and progesterone play an influential role in mammary gland development and lactation function regulation [17], and breast degeneration can be delayed as their concentration in the blood decreases. Near the time of delivery, the concentration of estrogen in the blood rises significantly, stimulating the pituitary gland to secrete PRL into the bloodstream and significantly increasing PRL receptors in the mammary cells, thus initiating lactation [18]. Additionally, estrogen exhibits a crucial effect in regulating the growth and development of breast tissue. Specifically, it stimulates the neuroendocrine growth axis and increases the level of somatotropin, thereby regulating the production of insulin-like growth factor (IGF-1) and the utilization of IGF-1-binding protein [19].

### 2.3. Bovine Somatotropin (BST)

Produced by the anterior pituitary gland, bovine somatotropin is a protein hormone that stimulates growth and affects a variety of metabolic processes. In addition to increasing the lipolytic activity of adipose tissue and inhibiting the transmission activity of glucose into tissue, it directly or indirectly stimulates growth processes, such as cell division, bone growth, and protein synthesis [20]. In recent years [21], somatotropin (ST) has been found to significantly affect lactation in ruminant animals, and the concentration of pituitary hormone is also directly related to lactation in cows. As reported by Gurevich et al., prepartum BST treatment significantly increases the plasma levels of hormones, insulin, and non-essential fatty acids (NEFA), while postpartum BST injections increase that of hormones and NEFA, but not IGF-1 [22] (Figure 1).

## 3. Regulation of Lactogenesis by Cytokines and Signaling Molecules in Dairy Cows

A number of cytokines and signaling molecules are involved in mammary gland remodeling, including the IGF family, TGF-β (transforming growth factor β), STAT, etc. The IGF-IGFBP (insulin-like growth factor-binding protein) system primarily regulates mammary gland remodeling during the dry milk phase, where IGF-I is up-regulated, resulting in an increase in cell renewal rate [23]. IGFBP-3 and IGFBP-5, synthesized and secreted by the mammary glands of dry milk cows, bind to IGF-I to inhibit its pro-proliferative signaling [24], thereby regulating mammary cell apoptosis.

The TGF-β family includes TGF-β1, TGF-β2, and TGF-β3 [25], among which TGF-β1 is the most extensively studied and is known to be involved in cell proliferation, cell differentiation, apoptosis, and extracellular matrix (ECM) synthesis. In particular, TGF-β1 regulates several genes to control the cell cycle process, including cell cycle regulatory proteins, cyclin-dependent protein kinases, cyclin-dependent protein kinase inhibitors, and proto-oncogenes [26]. It is also reported to regulate the apoptotic process. In addition, TGF-β1 is likely to alter the composition of the ECM, thus affecting mammary gland development [27]. Collagen genes are highly expressed when TGF-β1 is released from pills implanted into the interstitial fraction. TGF-β1, however, does not affect morphological changes or glandular follicle development during puberty [28]. There are three isoforms of TGF-β expressed at different times during cow development. The mRNA level of TGF-β1 is higher during puberty and lower during gestation and lactation, with the latter being the period of mammary gland remodeling when the mRNA level of TGF-β1 is decreased [29]. When TGF-β1 is slowly released into the mammary parenchyma, an increase in both epithelial cells and stromal cells can be observed, as can higher levels of fibronectin gene expression [30]. It is hypothesized that TGF-β1 induces an increase in the stroma and provides a stromal environment for epithelial cell growth, thus promoting epithelial cell proliferation. TGF-β2 and TGF-β3 affect the mammary gland in a similar manner [31].

STAT5, a member of a family of proteins, mediates the signaling of a range of cytokines and growth factors. Its DNA binding activity was originally identified in breast tissue [32]; hence, it was named the mammary gland factor (MGF). Recombinant STAT5 is capable of mediating the actions of several peptide hormones, including PRL, somatotropin (ST), erythropoietin (EPO), interleukin 3 (IL3), interleukin 5 (IL5), and granulocyte–macrophage colony-stimulating factor (Figure 2). Disruption of the STAT5 gene has been reported to impair mammary gland development and prevent lactation, suggesting that STAT5 is an important regulator of mammary gland development and lactogenesis function [33]. After delivery, in normally functioning mammary tissue, elevated levels of activated STAT5 are observed, while levels of phosphorylated STAT1 and STAT3 are reduced [34]. Therefore, STAT5 phosphorylation is deemed a key factor in the terminal undifferentiation of epithelial cells secreted by the mammary gland [35]. Furthermore, STAT5 plays an essential role in acidoid proliferation and differentiation and also serves as a key transcription factor for the whey acid protein gene [36] (Figure 2).

Sterol Regulatory Element Binding Transcription Factor 1 (SREBP1) is widely recognized as a significant mechanism facilitating fat synthesis by PUFAs. The results of qRCR studies have shown that there is a reduction in the levels of milk fat synthesis enzymes and suppression of their related transcription factors during milk fat inhibition. Conjugated linoleic acid (CLA) and transcription factors involved in regulating milk fat synthesis result in a synergistic reduction in lipogenic enzymes, particularly SREBP1, Peroxisome Proliferator-Activated Receptor γ (PPARG), etc. This provides compelling evidence that transcriptional regulation is the main mechanism by which CLA induces milk fat inhibition.

The augmented expression of SREBP1 in lactating cows and goats is thought to be associated with an upsurge in milk fat synthesis, primarily through its binding to sterol response elements located in the promoter region, which governs the transcription of its target genes that are mainly involved in lipogenesis. The transfection of SREBP1 siRNA in dairy cow mammary epithelial cells results in a decrease in the mRNA expression of acetyl-CoA carboxylase α (ACC), Fatty Acid Synthase (FAS), stearoyl-CoA desaturase-1 (SCD), and Fatty Acid-Binding Protein 3 (FABP3); it also causes a reduction in triglyceride secretion, which is partly dependent on the positive feedback effect of SREBP1 on the mTOR signaling pathway. Additionally, positive effects of SREBP1 on milk lipid synthesis have been demonstrated in goats.

CLA can reduce the expression of genes related to milk fat synthesis by inhibiting the activity and expression of SREBP1. The abundance of nuclear SREBP1 (nSREBP1) depends on the upstream SREBP cleavage-activating protein (SCAP) and Insulin-Induced Gene 1 (INSIG1). Additionally, trans10, cis12 CLA prevents the activation of SREBP1 by inhibiting the expression of SCAP and INSIG1, leading to a reduction in mRNA levels of the SREBP1 downstream genes ACACA (Acetyl-CoA Carboxylase α), FASN (Fatty Acid Synthase), and SCD1. In vitro studies have shown that although the transcription and translation levels of nSREBP1 remain unchanged after trans10, cis12 CLA treatment, the expression of SREBP1 mRNA is directly inhibited and reduced when 26S protease activity is inhibited.

The nuclear protein thyroid hormone response site 14 (THRSP, Spot14, S14) plays a crucial role in regulating milk lipid synthesis and exhibits high expression in the liver and adipose tissue and moderate expression in the mammary gland. In goat mammary epithelial cells, the overexpression of S14 is accompanied by the up-regulation of FASN, SCD1, GPAM (glycerol-3-phosphate acyltransferase, mitochondrial) expression, along with the down-regulation of CD36 (CD36 Molecule) expression, while ACACA levels remain unaffected. Furthermore, microarray analyses of bovine mammary tissue cultures have identified S14 as a candidate gene in response to trans10, cis12 CLA, similarly to SREBP1. The results of cDNA microarray analysis indicated down-regulation of S14 mRNA expression in bovine mammary tissue treated with trans10, cis12 CLA, although changes in protein levels were not examined. At the individual nucleotide level, the S14 promoter contains SRE and exhibits high sensitivity to nSREBP1. Therefore, S14 may function as a secondary cellular signal or transcriptional coactivator of SREBP1 and has been preliminarily demonstrated in transgenic mice. However, its validation in ruminants remains unexplored.

Another important transcription factor regulating ruminant milk fat metabolism, PPARG, is present in mammary tissue and has a facilitative effect on FASN, ACC, LPIN1, etc. It is noteworthy that C-terminal amino acid residues of LPIN1 spanning from 217 to 399 can activate PPARG. Surprisingly, the use of the peroxisome proliferator activated receptor γ (PPARγ) agonist does not stimulate lactating dairy sheep milk lipid synthesis, nor does it reverse the negative effects of trans10, cis12 CLA on the milk lipid production genes SREBP1, SCD1, and mTOR. In addition, trans10, cis12 CLA does not appear to elicit PPARG expression in bovine mammary epithelial cells [37]. Nevertheless, CLA influences milk fat synthesis through PPARs, and the transcriptional repression of milk fat synthesis genes by CLA is observed only upon basal activation of PPARG. In addition, PPARG has been shown to target INSIG, which is involved in the activation of SERBP1. We hypothesize that the repressive effect of CLA on SREBP expression might be counteracted when PPARG is repressed.

In conclusion, the regulatory mechanisms of CLA (mainly trans10, cis12 CLA) on the transcription factor SREBP1, which is involved in milk lipid synthesis, have been well elucidated. These mechanisms encompass a reduction in SREBP1 levels in the nucleus through the inhibition of upstream factors and the direct negative regulation of SREBP1 transcript levels under specific circumstances. However, the regulatory mechanisms of the other transcription factors, THISP and PPARG, in ruminant milk fat synthesis remain unclear. While S14 has been shown to impact the expression of PPARG and SREBP1, it does not appear to affect the expression of the downstream genes involved in lipid synthesis. The authors thus propose the existence of tautomers in ruminant mammary cells that function similarly to S14 and whose expression levels are down-regulated by trans10, cis12.

## 4. Regulation of Lactogenesis by Proteases in Dairy Cows

A variety of proteases are involved in the regulation of mammary gland remodeling [38]. Among them, fibrinogen is important in mammary gland development because fibrinogen-deficient mice show significant defects in lactogenesis capacity and mammary gland degeneration after lactogenesis [39]. Fibrinogen can be converted into a fibrinolytic enzyme by urokinase-type fibrinogen activator (uPA) [40], tissue-type fibrinogen activator (tPA), and plasma kinase-releasing enzyme. During mammary gland degeneration, tPA and uPA levels have been reported to increase dramatically, stimulating the activation of fibrinolytic enzymes [41]. Subsequently, activated fibrinolytic enzymes degrade αs-casein, β-casein, κ-casein, and lactoferrin, leading to changes in milk composition [42].

Mammary epithelial cells absorb fatty acids from plasma, including free fatty acids and those bound to lipoprotein. Lipoprotein esterase (LPL) can hydrolyze triglycerides in plasma and release fatty acids. Mammary epithelial cells absorb low-density lipoprotein and chylous long-chain fatty acids only after being digested by LPL. LPL is highly expressed in breast tissue, which accounts for its heightened activity in this region. During lactation, the expression of LPL undergoes sharp up-regulation in the early stage. In mice, the expression of LPL during lactation is only approximately two times higher than that during pregnancy, and the activity of LPL protease is also up-regulated approximately two-fold. During lactation, co-expression of LPL and VLDLR (very low-density lipoprotein receptor) is observed, with VLDLR also being up-regulated. These findings suggest that VLDLR plays a crucial role in activating LPL during lactation, and the expression of LPL is consistent with the up-regulation and down-regulation of LDL in blood. This mechanism is crucial for enabling mammary gland cells to absorb exogenous fatty acids from plasma.

Not all fatty acids in the blood enter breast cells directly through diffusion. Studies have shown that most fatty acids are dependent on protein-mediated active transport to enter breast cells. The fatty acid transport proteins CD36 and SLC27A (Solute Carrier Family 27 Member 1) are crucial for the transport of fatty acids in non-ruminant cells. In ruminant mammary gland cells, CD36 is thought to be related to the secretion of cream droplets because of its presence on the adventitia, which is coated with cream droplets. As an increasing number of studies have shown, similar to non-ruminants, CD36 also plays an indispensable role in the absorption of fatty acids in the mammary gland cells of ruminants. Furthermore, multiple SLC27A subtypes have been revealed to be expressed in the mammary glands of dairy cows, but only SLC27A6 is up-regulated in the first stage of lactation. CD36 is also frequently co-located with transport proteins, such as acid acetyl-CoA synthase family (ACSL) and FABP, and its up-regulation and that of SLC27A increase their ability to anchor to the cell membrane.

Upon entering breast cells, long-chain fatty acids require activation through increased COA before they can participate in the physiological metabolic process. The activation of fatty acids is mainly accomplished by ACSL. In the ACSL family, ACSL1 is most abundant in breast tissue and is up-regulated more than four-fold during the lactation cycle, suggesting that it has a significant role in milk fat metabolism. Among the enzymes involved in the activation of short-chain fatty acids in dairy cow mammary glands, the expression of acetyl-CoA short-chain fatty acid family 2 (ACSS2) is the highest, with a higher up-regulation ratio than that of ACSS1. In mice, ACSS family amino acids exhibit a similarity rate of approximately 43.8% and are distributed across various cell types. While both enzymes exhibit a high affinity towards acetate, ACSS2 displays a stronger affinity. ACSS1 has a robust affinity towards propionate. Functionally, it is inclined to deoxidize acetic acid, while ACSS2 prefers the involvement of activated acetic acid in the re-synthesis of fatty acids.

Due to the low efficiency of the free diffusion of fatty acids in cells and the absence of biological targeting, most targeted transport of fatty acids in cells necessitates the participation of relevant transport proteins. In non-ruminants, FABP and acetyl-CoA binding protein (ACBP) are primary intracellular fatty acid transport proteins. Although ACBP is expressed in both bovine and mouse mammary glands, its expression is relatively low, leading to speculation that its function in mammary glands may not be significant. On the contrary, the FABP family exhibits high expression in mammary glands, especially FABP3. In fact, the expression of FABP3 in bovine mammary glands is the highest among all subtypes and is up-regulated approximately 80-fold during lactation. Although FABP4 and FABP5 exhibit up-regulation during lactation, their levels are comparatively lower than that of FABP3. In addition to its transport function, FABP shows the capacity to influence the activities of ACACA and SCD by binding to activated acetyl-CoA. Given that FABP facilitates the transport of fatty acids for SCD desaturation, it can be inferred that in breast tissue, FABP3 is responsible for transporting substrates such as C16 or C18 to desaturate SCD, and then transporting the desaturated fatty acids to FABP4.

In mammary tissue, MEC and mesenchymal tissue are separated by the extracellular matrix [43], which is responsible for maintaining the differentiated state of the mammary gland and regulating apoptosis. During mammary gland degeneration, the activation of the matrix metalloproteinase (MMP) system induces the protein hydrolysis of extracellular matrix components, leading to extensive mammary cell apoptosis and degeneration [44]. In addition, MMPs release growth factors and cytokines that are involved in the regulation of other physiological processes during dry lactation, such as activation of the immune system and cell growth [44].

## 5. Regulation of Lactogenesis by Other Factors in Dairy Cows

In addition to physiological factors such as self-hormones, proteases, and cytokines, pathological and environmental factors, such as photoperiod [45], nutrition level [46], heat stress [47], and oxidative stress [48], also affect mammary remodeling during dry lactation [49]. The length of the dry milk photoperiod influences mammary remodeling and its subsequent lactation cycle. Compared to long-day photoperiods, short-day photoperiods result in higher milk production and improve immune function in the next lactation cycle [50]. Additionally, short-day photoperiods have been found to regulate mammary gland remodeling during dry lactation by promoting mammary cell proliferation and reducing MEC apoptosis. Heat stress affects the morphology of mammary cells and processes such as apoptosis and proliferation [51], and also decreases protein synthesis in mammary cells, leading to apoptosis and autophagy.

Light duration influences the regulation of lactogenesis in dairy cows, as well as in other ruminants. It has been reported that milk production in lactating cows also differs significantly depending on weaning time. Additionally, light affects both milk production and the number and activity of mammary epithelial cells. The duration of light exposure also influences the content of hormones (e.g., PRL, melatonin, and IGF-1) in the organism. Therefore, the role of light in the regulation of lactogenesis cannot be underestimated [52] (Figure 3).

## 6. Conclusions

As the dairy industry has developed rapidly and people’s living standards have improved substantially, the consumer market for dairy products has also expanded. Presently, research is focused on how to improve milk production and optimize milk quality in the industry [53]. The mammary gland serves as the lactation organ in dairy cows, with bovine mammary epithelial cells (BMECs) acting as the basic functional unit for breast milk synthesis. During lactation, lactation-related hormones and cytokines play an indispensable role in regulating the development of mammary epithelial cells and lactation in dairy cows. In recent years, the study of BMECs has involved the incorporation of various lactation-related hormones and cytokines. A large number of studies have revealed that different lactation-related hormones and cytokines have distinct effects on the proliferation and expression of lactation-related genes in BMECs. Additionally, alterations in concentration lead to dynamic changes in their expression levels, underscoring the pivotal role of lactation-related hormones and cytokines in the regulation of BMEC lactation. Therefore, a wealth of information has been obtained by tracing the lactation initiation process and by analyzing and discussing the hormones, proteases, cytokines, and signaling molecules involved, as well as other related factors, on lactation initiation and the lactation process in dairy cows. This analysis suggests that deeper comprehension of the roles and regulatory mechanisms of key factors in different lactation periods in dairy cows could lead to improved milk yield and quality [54]. In this review, we show that the study of how different key factors interact with each other, such as the collaborative actions of hormones and proteases in lactation regulation, remains an unexplored but potentially valuable and significant topic for future milk research [55]. In conclusion, revealing the functions of key factors and their synergistic mechanisms will be the focus of future research in this field. These studies will shed new light on the formation mechanism of mammary gland-initiated lactation in dairy cows and provide worthwhile theoretical evidence for the regulation of milk fat composition.

## Figures and Tables

**Figure 1 genes-14-01163-f001:**
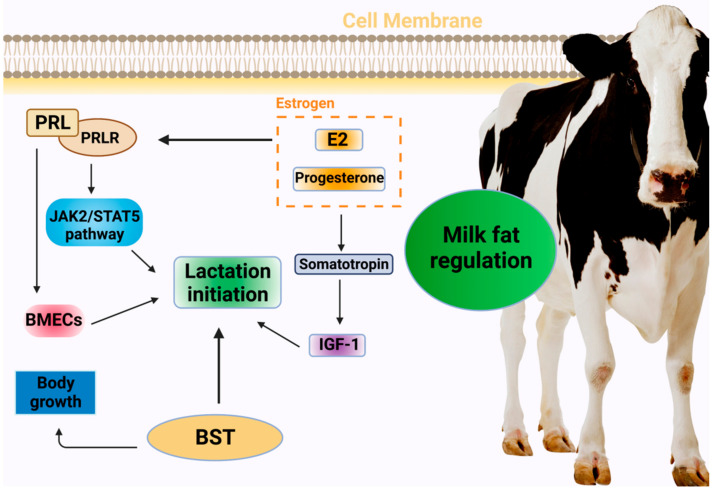
Hormonal regulation of milk initiation in dairy cows. PRL: prolactin; PRLR: prolactin receptor; JAK2: T Janus Kinase 2; STAT5: signal transducer and activator of transcription 5; BMECs: bovine mammary epithelial cells; E2: estradiol; BST: bovine somatotropin; IGF-1: insulin-like growth factor.

**Figure 2 genes-14-01163-f002:**
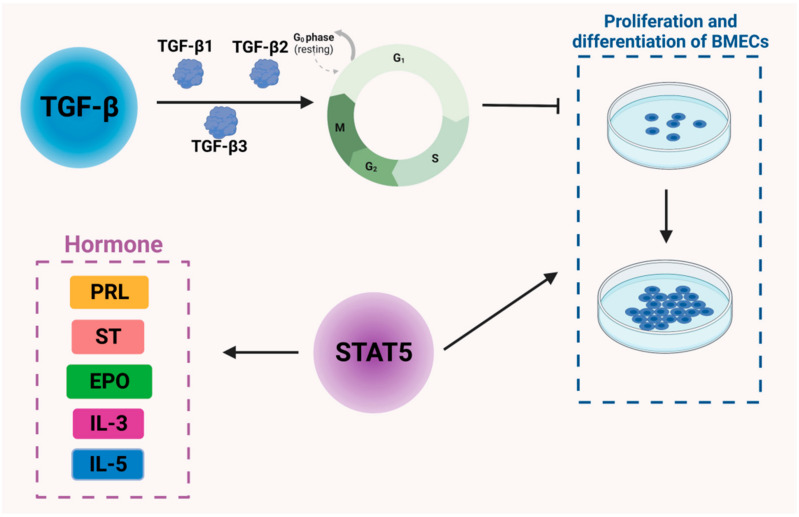
Regulation of cytokines and signaling molecules in bovine mammary epithelial cells and mammary gland development in dairy cows. TGF-β: transforming growth factor β; PRL: prolactin; ST: somatotropin, EPO: erythropoietin, IL3: interleukin 3, IL5: interleukin 5.

**Figure 3 genes-14-01163-f003:**
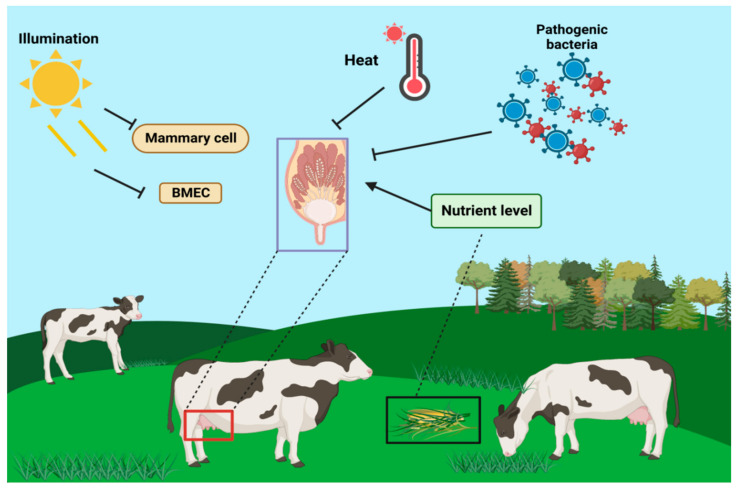
Effects of light cycle, nutrient level, heat stress, pathologic factors, and environmental factors on mammary gland tissues of dairy cows.

## Data Availability

Not applicable.

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
