# Peer review of "Progress in Research on Key Factors Regulating Lactation Initiation in the Mammary Glands of Dairy Cows"

_genes, 2023, doi:10.3390/genes14061163_

Round 1

Reviewer 1 Report

-The paper presents some interesting information about the endocrine control of the development of the mammary gland, and the initiation of lactation.

-The review in Section 1 presents established endocrine mechanisms, and I suggest that the important hormones be summarized in a more concise and shorter way in this section.

-The real new information is in Sections 2, 3 and 4, which deserves a more detailed review, for this paper to make a meaningful contribution to this field of study. 

- Some old references are use and should be replaced to truly quality for "new findings".

-The general English use is poor - my impression is that it was written to save space, but it makes reading more difficult (see suggestions in PDF version of paper).

The general English use is poor - my impression is that it was written to save space, but it makes reading more difficult (see suggestions in PDF version of paper).

Author Response

The paper presents some interesting information about the endocrine control of the development of the mammary gland, and the initiation of lactation.

Response: Thank you for your valuable advice, and we will solve and answer your questions one by one.

-The review in Section 1 presents established endocrine mechanisms, and I suggest that the important hormones be summarized in a more concise and shorter way in this section.

-The real new information is in Sections 2, 3 and 4, which deserves a more detailed review, for this paper to make a meaningful contribution to this field of study.

Response: Thank you for your meaningful suggestion. We have made modifications to the section 1 (introduction). This article is not just a review of hormones. Mainly divided into the following points:

  1. Hormonal regulation of lactation in dairy cows

2 Regulation of lactogenesis by cytokines and signaling molecules in dairy cows

3 Regulation of lactogenesis by proteases in dairy cows

4 Regulation of lactogenesis by other factors in dairy cows

Our detailed content is in sections 1-4 above.

- Some old references are use and should be replaced to truly quality for "new findings".

Response: Another reviewer suggested supplementing some earlier literature. We have further revised the references.

-The general English use is poor - my impression is that it was written to save space, but it makes reading more difficult (see suggestions in PDF version of paper).

Response:

a. For the language, we have carefully revised it. We also invited the polishing company recommended (https://www.mdpi.com/authors/english) by the magazine to make language modifications (ID: 66399).

b. We did not find the PDF version of paper.

Reviewer 2 Report

The manuscript by Guo et al. analyzed Progress of research on key factors regulating lactation initiation in the mammary gland of dairy cows. These studies will be shed new light on elucidating the formation mechanism of mammary gland-initiated lactation in dairy cows and provide important theoretical support for the regulation of milk fat composition. It is a topic of interest to the researchers in the related areas but the paper needs improvement before acceptance for publication. The following are the questions and mistakes in this manuscript.

1.What is the range of lactation-related hormone levels in cows at different lactation periods?

2.As we all know, dairy cattle include Holstein cattle, Jersey cattle, buffalo, etc., what are the differences and similarities between their mammary gland lactation regulation?

3.What is the meaning of "kinol" in line 65, “···kinol are pivotal to the efficiency of substance metabolism, transformation and uptake···”?

4.Please list all of the authors of the last reference.

Author Response

1.What is the range of lactation-related hormone levels in cows at different lactation periods?

Response: Thank you for your question. The content of reproductive hormones in milk cow plasma is as follows:

Items

Levels

PRL(prolactin,μIU/mL)

153.38±7.19

P(progesterone,pg/mL)

386.67±37.08

T(testosterone,pg/mL)

59.38±4.42

E2(estrone,pg/mL)

542.33±49.01

PGE2(prostaglandin E2,pg/mL)

38.84±4.24

OT(oxytocin,pg/mL)

27.61±2.87

2.As we all know, dairy cattle include Holstein cattle, Jersey cattle, buffalo, etc., what are the differences and similarities between their mammary gland lactation regulation?

Response: Holstein cow, Jersey cow, and buffalo milks were similar in FA profiles and rich in ECSFA. Yak milks was rich in OBCFA and n-3 PUFA, and considered as potential functional foods for balanced human diet. Higher expression level of genes related with fat synthesis including FA de novo synthesis related genes (FAS and ACACA), Glyceride synthesis related genes (GPAT and AGPAT6), and transcriptional regulator related genes (SREBP1 and PPARG) in buffalo as compared to Holstein cow milk.

3.What is the meaning of "kinol" in line 65, “···kinol are pivotal to the efficiency of substance metabolism, transformation and uptake···”?

Response:  Modified.

  1. Please list all of the authors of the last reference.

Response:  Modified.

Reviewer 3 Report

Concerns

1.    Earlier literature has been ignored. 

2.    Are there not roles for thyroid hormones and cortisol in initiation of lactation.

3.    The word, lactogenesis, is not used. 

4.    There is a lack of consistency of abbreviations and terminology

For instance, it is not clear why the authors have abbreviated growth hormone as bST.  After first usage of a hormone, abbreviations should be used throughout.

5.    Figure 1

·      Inconsistent case (upper and lower) for “MiLK”.

·      Not sure what “Hormone” signifies.

·      Why are “somatotropin” and “BST” exerting separate effects?

·      Define all abbreviations in the figure legend e.g. BMEC, E2

6.    Figure 2

·      Define all abbreviations in the figure legend e.g. BMEC.

·      Where is the evidence that EPO, IL-3 and IL-5 influence proliferation of BMEC cells?

The manuscript is difficult to understand. It really needs a Native English speaker or editing service to go the manuscript carefully. 

Author Response

  1. Earlier literature has been ignored. 

Response: We have supplemented early literature reports.

  1. Are there not roles for thyroid hormones and cortisol in initiation of lactation.

Response:

       a. Research reports have shown that there is a certain relationship between thyroid hormones and prolactin, but it does not directly regulate lactation (J Hiroi; 1997).

      b. Thyroid hormone is necessary for fetal metabolism and growth. Fetal thyroid hormone affects the growth and development of the fetus to a large extent, leading to the occurrence of low birth weight infants. But it is not related to the initiation and maintenance of lactation.

     c. Cortisol is a hormone secreted by the adrenal gland, also known as cortisol. It is a type of steroid hormone with multiple biological effects. Cortisol plays a very important role in regulating metabolism, immunity, cardiovascular system, central nervous system, and other aspects of the body. It is not related to the initiation and maintenance of lactation (J Gaab; 2005).

Reference

(1)J Hiroi,Y Sakakura,M Tagawa,T Seikai,M Tanaka.Developmental Changes in Low-Salinity Tolerance and Responses of Prolactin, Cortisol and Thyroid Hormones to Low-Salinity Environment in Larvae and Juveniles of Japanese Flounder, Paralichthys olivaceus. Zoologicalence, 1997, 10.2108/zsj.14.987

(2)J Gaab,N Rohleder,UM Nater,U Ehlert. Psychological determinants of the cortisol stress response: the role of anticipatory cognitive appraisal. Psychoneuroendocrinolog, 2005, 10.1016/j.psyneuen.2005.02.001

  1. The word, lactogenesis, is not used. 

Response: Lactogenesis is the onset of milk secretion and includes all of the changes in the mammary epithelium necessary to go from the undifferentiated mammary gland in early pregnancy to full lactation sometime after parturition. We have supplemented and revised it in the article.

  1. There is a lack of consistency of abbreviations and terminology

For instance, it is not clear why the authors have abbreviated growth hormone as bST.  After first usage of a hormone, abbreviations should be used throughout.

Response: Thank you for your meaningful suggestion. We have made modifications throughout the entire text.

  1. Figure 1
  • Inconsistent case (upper and lower) for “MiLK”.

  Response: Modified

  • Not sure what “Hormone” signifies.

Response: Deleted

  • Why are “somatotropin” and “BST” exerting separate effects?

  Response: We uniformly use 'somatotropin' instead of growth hormone

  • Define all abbreviations in the figure legend e.g. BMEC, E2

Response: We have made modifications and supplements in the article

  1. Figure 2
  • Define all abbreviations in the figure legend e.g. BMEC.

Response: We have made modifications and supplements in the article

  • Where is the evidence that EPO, IL-3 and IL-5 influence proliferation of BMEC cells?

Response:

  1. Recombinant STAT5 is capable of mediating the actions of several peptide hormones, including prolactin.
  2. STAT5 is essential for acinoid proliferation and differentiation and is a key transcription factor for the whey acid protein gene.
  3. We have made moderate modifications to the figures in the article.

Comments on the Quality of English Language

The manuscript is difficult to understand. It really needs a Native English speaker or editing service to go the manuscript carefully. 

Response: For the language, we have carefully revised it. We also invited the polishing company recommended (https://www.mdpi.com/authors/english) by the magazine to make language modifications (ID: 66399).

Round 2

Reviewer 3 Report

The authors have satisfactorily address all my concerns.